# Spatiotemporal forelimb muscle activation during precise asymmetric stepping in rats

Kacie Hanna[1⊙], Ezequiel M. Salido[2,3⊙], Neha Lal[4], Kiril Tuntevski[4],
Sergiy Yakovenko[1,4,5,6,7*]

1 Department of Biomedical Engineering, Benjamin M. Statler College of Engineering and Mineral Resources, West Virginia University, Morgantown, West Virginia, United States of America, 2 Department of Biochemistry and Molecular Medicine, School of Medicine, West Virginia University, Morgantown, West Virginia, United States of America, 3 Department of Ophthalmology and Visual Science, School of Medicine, West Virginia University, Morgantown, West Virginia, United States of America, 4 Department of Human Performance – Exercise Physiology, School of Medicine, West Virginia University, Morgantown, West Virginia, United States of America, 5 Rockefeller Neuroscience Institute, School of Medicine, West Virginia University, Morgantown, West Virginia, United States of America, 6 Mechanical and Aerospace Engineering, Benjamin M. Statler College of Engineering and Mineral Resources, West Virginia University, Morgantown, West Virginia, United States of America, 7 Department of Neuroscience, School of Medicine, West Virginia University, Morgantown, West Virginia, United States of America

⊙ Authors contributed equally to this study
* seyakovenko@mix.wvu.edu

## Abstract

A sequence of muscle actions generates complex movements such as walking or reaching. However, how these coordinated actions subserve complex movements across animals remains unknown. While the sequences of muscle activity have been documented in limb tasks with large animals, the equivalent comprehensive behavioral description of rodent performance is sparse. To this end, we have trained rats to perform precise foot placement, which allows us to assess skilled limb placement during locomotion. Animals were trained on the pegway task, conFigd to impose symmetric or asymmetric (with overstepping) locomotor stepping at the preferred stride length. We collected electromyography from selected representative forelimb muscles implanted with intramuscular differential electrodes and recorded ground reaction forces from the array of force sensors embedded into walkway pegs. The changes in muscle coordination were analyzed for symmetric and asymmetric stepping. The sequence corresponded to the progression of muscle actions responsible for limb lift, flexion and transport, overground clearance, and preparation for ground contact. The stereotyped spatiotemporal sequence of muscle activity was persistent and consistent across asymmetric tasks. These patterns are similar to those observed in cats during locomotion over obstacles and reaching movements. These findings indicate that a temporal sequence of muscle actions is similar across quadrupeds during locomotor tasks with fine stepping control.

**Data availability statement:** All data and analysis code underlying the results are publicly available without restriction. We deposited data and MATLAB scripts in an open repository: 10.6084/m9.figshare.29915057 / https://figshare.com/s/c46e925d2b68b85311cc.

**Funding:** NIH/NIGMS P20GM109098 (SY, ES, NL), NIH/NIGMS U54GM104942 (KT), NIH/NIGMS T32 GM132494 and NIH/NIA T32 AG052375 (KH). The funders played no role in the study design, data collection and analysis, decision to publish, or preparation of the manuscript.

**Competing interests:** The authors have declared that no competing interests exist.

## Introduction

Locomotion is essential to the survival of all animals [1]. From the flagellar movement of unicellular organisms to the limbed locomotion generated by coordinated muscles of mammals, mobility offers distinct evolutionary advantages in finding prey, escaping predators, and courting mates. Yet, it is often challenging to navigate unpredictable terrain with obstacles and dangers requiring an animal to change direction or speed rapidly.

In vertebrates, the neural structures involved in locomotion are incorporated into a complex hierarchical system that seamlessly integrates commands from supraspinal motor pathways, intrinsic spinal rhythmogenic networks, and afferent sensory feedback [2,3]. At the top of the motor hierarchy, the motor cortex has been shown to be an essential contributor for the accurate spatiotemporal control of discrete volitional limb movements, such as reaching for a target in both cats and primates [4–9], as well as for real-time modification of stereotypical rhythmic movements such as locomotion [10–17] and mastication [18,19]. In addition, single-unit recordings during such tasks have identified active pyramidal tract neurons (PTNs) that fire in specific patterns related to the change in task, such as during readjustment of reach trajectory [20,21], when changing gait length or stepping over an unexpected obstacle [10,12,14,20,22,23], and during changes based on visual [22] or proprioceptive feedback [24]. Moreover, these units fire discretely in relation to specific muscle groups during these tasks [20,23], supporting the idea of control modules expressed within the nervous system.

The organization of downstream control elements has also been shown to express modularity, which was observed as the task-specific grouping of muscles in stereotypical behaviors such as locomotion, scratching, and swimming [25–31]. Sequential muscle group patterns are also observed in postural control of cats and humans [21,32], in locomotion of cats and humans [33–35], and in reaching movements of cats and primates, including humans [14,20,36–38]. The existence of downstream modularity suggests that this organizational principle offers advantages for integrating contributions from multiple neural pathways.

Yet, despite significant redundancy built into the motor control system, both central injuries and peripheral trauma can disrupt motor function. Cerebrovascular accidents or strokes are among the primary causes of adult disability [39–41], and can cause direct damage to the motor cortex. In hemiparetic patients, this results in an asymmetric gait [42–45], which may persist even after rehabilitation therapy [46–49]. An interesting aspect of this hemiparetic gait is the heterogeneity within the patient population; either the paretic leg or the non-paretic leg may exhibit a shorter stride [43,45]. The lateralization of deficits in the hemiparetic gait can be characterized by the spatial and temporal asymmetry of step length [50], and may improve assessment and therapeutic targets in locomotor rehabilitation [51,52]. Another assessment method is the analysis of muscle activity profiles; merging and fracturing of temporal profiles in muscle groups with shared sources of activity (also called synergies) describe deficits in post-stroke patients [53–57]. Further detailed characterization of the asymmetric locomotion in standard animal models could facilitate a more granular understanding of muscle coordination and motor modules in health and disease.

This rationale-driven study documents the coordination of muscle templates in rats, a widely used animal model for injury research and the development of therapeutics. Firstly, rats are well-defined models of stroke and spinal cord injury [58]. Secondly, they have lissencephaly, or a smooth brain surface, which is appropriate for the use of high-density electrode arrays. Finally, rats are also stereotaxically consistent animals, making them suitable for investigating cortical and subcortical networks [59]. However, rats are small quadrupedal animals that have not been neurophysiologically characterized to the same extent as cats, monkeys, or humans. There are clear similarities between the gait patterns of cats and rats, but notable differences also exist. For example, rats prefer to run or trot with a crouched posture, bearing a greater proportion of their weight on the hind legs [60]. Rats exhibit freezing in the presence of predator odors [61] and standing up driven by curiosity [62]. The argument could be made that these differences do not guarantee that the neuromechanical dynamic system operates in the same regime. Therefore, there is a need to characterize locomotor behaviors and test the equivalence or investigate the disparity between cats and rats at both the behavioral and structural levels. Closing this gap is the essential step toward establishing the rat as a model to further explore the descending control of muscle coordination during normal and pathological conditions. Here, we determined the spatiotemporal progression of forelimb muscle activity during symmetric and asymmetric stepping in rats and tested the hypothesis that their activation sequence parallels that of cats.

## Methods

The experiments were carried out in eight adult female Sprague-Dawley rats (225-275 g, 2–3 months old). Female rats were used for convenience and consistency when combined with the injury model, such as medial cerebral artery occlusion [63]. A set of representative forelimb muscles was implanted with differential intramuscular electrodes (see details below) either bilaterally or unilaterally, with a representative muscle on the contralateral side. The unilateral EMG implants were necessary to accommodate a cortical 16-channel FMA implant. Together, we collected data from the representative muscle groups of 13 limbs. The animals were trained on the pegway task for one week prior to the surgical procedures. All experimental procedures involving animals were reviewed and approved by the Institutional Animal Care and Use Committee (IACUC) of West Virginia University (Protocol #15–0303). All methods were carried out in accordance with the National Institutes of Health Guide for the Care and Use of Laboratory Animals and in compliance with relevant institutional, state, and federal regulations. The study did not involve human participants; therefore, informed consent was neither required nor obtained.

## Surgery

All surgical procedures were performed under general anesthesia and under aseptic conditions. Anesthesia was induced with 5% isoflurane and maintained with 1–3% isoflurane and oxygen. A pre-emptive analgesic, Buprenorphine SR, was administered subcutaneously at 1.2 mg/kg, along with Dexamethasone at 2 mg/kg to reduce inflammation. For acute pain management and vasoconstriction, Lidocaine HCl at 4 mg/kg was injected at all incision sites prior to making incisions. The surface of the eyes was coated with petroleum jelly to prevent drying out. As described previously for mice [64] and cats [3,21,65], forelimb muscles were implanted with pairs of Teflon-insulated, braided stainless steel wires (AS 633; Cooner Wire, Chatsworth, CA) for intramuscular electromyography (EMG). The following representative muscles of the forelimb were chosen: limb retractors—*latissimus dorsi* (LtD) and *spinodeltoid* (SpD); limb protractor flexing shoulder and elbow—*cleidobrachialis* (ClB); elbow flexor and wrist dorsiflexor transferring the limb during swing—*extensor carpi radialis longus* (ECR); wrist extensor and wrist plantarflexor preparing the limb for the ground contact and the transition to stance—*extensor digitorum communis* (EDC) and *palmaris longus* (Pal); and elbow extensors preparing limb for load-bearing during the onset of stance—*triceps brachii longus* (TrLo) with *triceps brachii lateralis* (TrLa).

The rats were placed in a stereotaxic apparatus. Four stainless steel self-tapping anchor screws (FST, Foster City, CA) were set at a depth of 0.25 mm in the cranium, approximately 3−4 mm anterior and 2−3 mm posterior to the lambda

fissure, and 3−4 mm from the midline (note: the exact location of the screws was adjusted to accommodate individual animal morphology). Two 18-pin Omnetics connectors (A79038-001, MicroProbes, Gaithersburg, MD) originating from the EMG implants were passed into custom 3D-printed housing, which was secured to the anchor screws with dental acrylic. This assembly of two connectors was attached to a single 32-channel interface board that directly plugged into a digitizing frontend (nano2stim+, Ripple, Salt Lake City, UT), allowing communication with the recording equipment (Grapevine Neural Interface Processor). Buprenorphine SR (1.2 mg/kg, SQ) was administered prior to the surgery and provided post-surgery analgesia for 72 hours. Ketoprofen (3 mg/kg) was administered as necessary to reduce any discomfort during the recovery period. Antibiotics were administered prophylactically for 5 days after surgery.

## Behavioral task

Animals performed locomotor tasks on an adjustable 24-peg walkway instrumented with an array of force-sensitive resistors embedded into pegs to record vertical ground reaction forces from foot-strike (**Fig 1**, $F_v$) [66]. The position of pegs could be modified to impose either a symmetric or asymmetric gait. We empirically determined 15 cm to be the preferred stride length for the rats by speeding up and slowing down locomotion on a treadmill until animals performed a continuous behavior without stopping or moving to the front of the treadmill (Exer-3/6 Treadmill, Columbus Instruments, OH). In general, animals change continuously gait speed in overground locomotion [67,68]; the speed over pegs was maintained through training to be within the target walking speed that was not interrupted by stopping or bouts of running. Typically, rats were easily trained to perform the peg task and often required mostly verbal encouragement to perform it for over 100 steps.

Symmetric and asymmetric gait types were represented by three conditions shown in **Fig 1B**. The first symmetric condition, S15 (black), corresponded to regular stride length symmetry with equal and out of phase stepping imposed by pegs set 7.5 cm apart, see **Fig 1B** (middle). The second and third conditions impose asymmetric gait with either *ipsilateral* or *contralateral* limb preference imposed by the asymmetric placement of pegs 9 and 6 cm apart (**Fig 1B** I9C6=red and I6C9=blue conditions). These stride conditions were alternated, and analyses were done relative to the EMG implantation

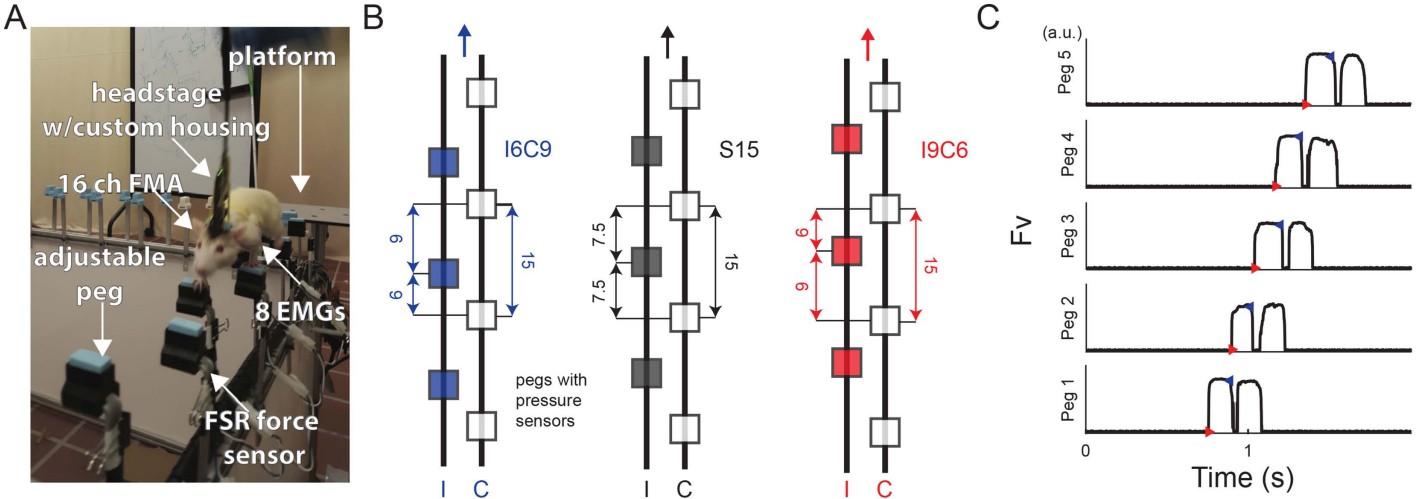

**Fig 1. Behavioral task. (A)** Implanted animal performing an asymmetric gait task on pegway. **(B)** Diagram of peg placement in symmetric (*S15* in black) and ipsi- and contra- lateral preferred asymmetric gait conditions (*I6C9* in blue and *I9C6* in red). **(C)**: Examples of vertical ground reaction force (*Fv*) from consecutive pegs. The events corresponding to the onset of stance (red) and swing (blue) of forelimbs were marked with an automated supervised method.

side. **Fig 1C** shows two contacts on each peg. The second contact was performed by the ipsilateral hindlimb and removed from analyses in subsequent figures. Animals were rewarded for the successful completion of each trial with verbal encouragement and food (Frootloops, Kellogg's, Battle Creek, MI).

## Data collection

After a week of recovery following the implantation, animals performed the locomotor tasks while tethered to the acquisition system (Grapevine Neural Interface Processor, Ripple, Salt Lake City, UT). EMGs and ground reaction forces were recorded simultaneously. Single-ended recordings from the implanted pairs of electrodes were conditioned and sampled at 30 kHz using a lightweight digitizing frontend. The EMG sampling was high to accommodate for the simultaneous cortical unit recordings, and it was downsampled to 1 kHz for temporal sequence analyses. The synchronized ground reaction forces (GRF) generated by contacts of both forelimbs and hindlimbs with each peg were conditioned and sampled at 1 kHz with an analog-to-digital card frontend. The dataset included only consecutive steps where the animal walked continuously without pausing or running.

## Signal processing

Digitized vertical forces ($F_v$) recorded with an array of force-sensing resistors were analyzed using a custom script in MATLAB (MathWorks, Natick, MA). The time of stance onset (red triangles) for forelimbs was semi-automatically marked using a supervised onset detection method [69] for all steps in each trial (**Fig 1C**). Only behavior with at least three consistent uninterrupted steps identified by the standard deviation of cycle duration (less than mean±2 st. dev.) was included in the analyses. The initiation of the transport phase can be detected by the onset of the deflection in the force profile, limb transport event in **Fig 2**, and stance terminations marked manually.

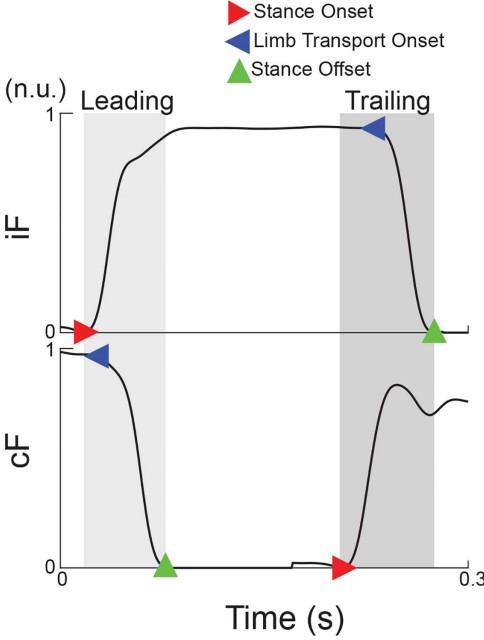

**Fig 2. Example of force signals identifying double stance leading and trailing phases.** Ipsilateral limb ground reaction force (*iF*) and contralateral ground reaction force (*cF*). Leading double stance time is shown in light gray shaded area between *iF* stance onset (red triangle marker) and *cF* stance offset (green triangle marker). Similarly, trailing double stance time is shown in the dark gray shaded area between *cF* stance onset and *iF* stance offset.

The quality of raw EMG patterns in a representative trial example is demonstrated in **Fig 3**, providing support for the analysis of signals in sessions. The EMG envelope profiles were created by high-pass filtering, rectifying, and low-pass filtering (high-pass second-order Butterworth filter cutoff frequency of 20 Hz and low-pass second-order Butterworth filter cutoff frequency of 100 Hz) using custom scripts (MATLAB, MathWorks, Natick, MA). Each EMG signal was normalized to the peak-to-peak value of the corresponding symmetric walking condition and averaged across all steps in the session. Analyses include 129±40 steps per animal.

## Results

### Intralimb coordination

During the walking on pegs, animals executed precise limb placements to step from rung to rung. Therefore, to make the comparison between progression of muscle activity in the cat during reaching [20,21] and the rat during this task, we analyzed ipsilateral muscle burst latency during limb transfer phase. In this study, transport phase was defined as the period between the onset of limb transport and the onset of ground contact. **Fig 2** illustrates the position of the onset of limb transport event (blue marker) relative to the stance offset (green marker). This onset event was associated with the onset of proximal limb muscles generating limb retraction during swing initiation. **Fig 3** shows examples of signals recorded during the three consecutive steps on the pegs. While the onsets of activity are generally clear and correspond to the progression of muscle actions, the automatic detection of these events is a challenging task. The valid alternative is the analysis of averaged signals within a session. For this purpose, the EMG signals were aligned with the onset of ipsilateral limb transfer and normalized to the duration of the ipsilateral transfer phase. Schematic stick Figs that illustrate phases of the step cycle are included in **Figs 4** and **5** to aid in visualization of results.

The muscle activity profile follows a standard sequence indicated by the increasing onset latency. **Fig 4A** shows representative signal averages in one subject. The onset latency of muscle activity was expressed in units normalized to the phase of limb transport. The onsets of muscle activity following the onset of limb unloading were manually identified (see arrows in **Fig 4A**). The superimposed raw signals were used as a reference to mitigate the adverse effects of low-pass filtering on the detection of events [70]. Signals were excluded from analysis if there was evident electrode failure. On average, data from eleven limbs were available per muscle.

During both symmetric (**Fig 4B**) and asymmetric (**Fig 4C**) stepping tasks, limb retractors *LtD* and *SpD* are the first to activate and lift the limb off the ground. Then, limb protractors, represented by *CIB*, and *ECR* transport the limb in swing. Wrist extensors represented by *EDC* are next to help with toe clearance over ground. Finally, wrist plantar flexor *Pal* and elbow extensors *TrLo* and *TrLa* prepare for foot contact and limb loading in stance (**Table 1**). The described muscle progression is used to accomplish the lift, flexion and transport, toe clearance, and prepare-for-contact stages of swing phase.

To test whether onset timing differed across muscles, we performed a repeated-measures analysis of variance (RM-ANOVA) with *Muscle* as a within-subject factor (8 levels). The analysis revealed a significant main effect of *Muscle*, $F(7, 14) = 28.55$, $p < 0.001$, Greenhouse–Geisser corrected $p = 0.01$, indicating that onset latencies varied systematically among the eight muscles. The effect was large, with partial eta-squared $= 0.94$.

To evaluate whether EMG onset latencies followed the hypothesized sequential progression, muscles were organized into four functional groups: G1 (*LtD*, *SpD*), G2 (*CIB*, *ECR*), G3 (*EDC*), and G4 (*Pal*, *TrLo*, *TrLa*), described in our previous work in cats [71]. Expecting the same sequence observed in cats, we used one-sided paired *t*-tests for the three successive comparisons, with the Bonferroni adjustment to maintain an overall $\alpha = 0.05$ across the sequence. All three planned comparisons were significant: G1 < G2: $t(12) = -5.41$, $p < 0.001$, Cohen's dz $= -1.50$; G2 < G3: $t(12) = -5.69$, $p < 0.001$, Cohen's dz $= -1.58$; G3 < G4: $t(12) = -7.63$, $p < 0.001$, Cohen's dz $= -2.12$, marked with (*) in **Fig 4B**. This supports the hypothesis that the sequence of muscle activity is maintained across symmetric and asymmetric stepping.

The analysis for two asymmetric patterns of walking on pegs is repeated and shown in **Fig 4C**. The same temporal progression was observed between four tested muscle groups.

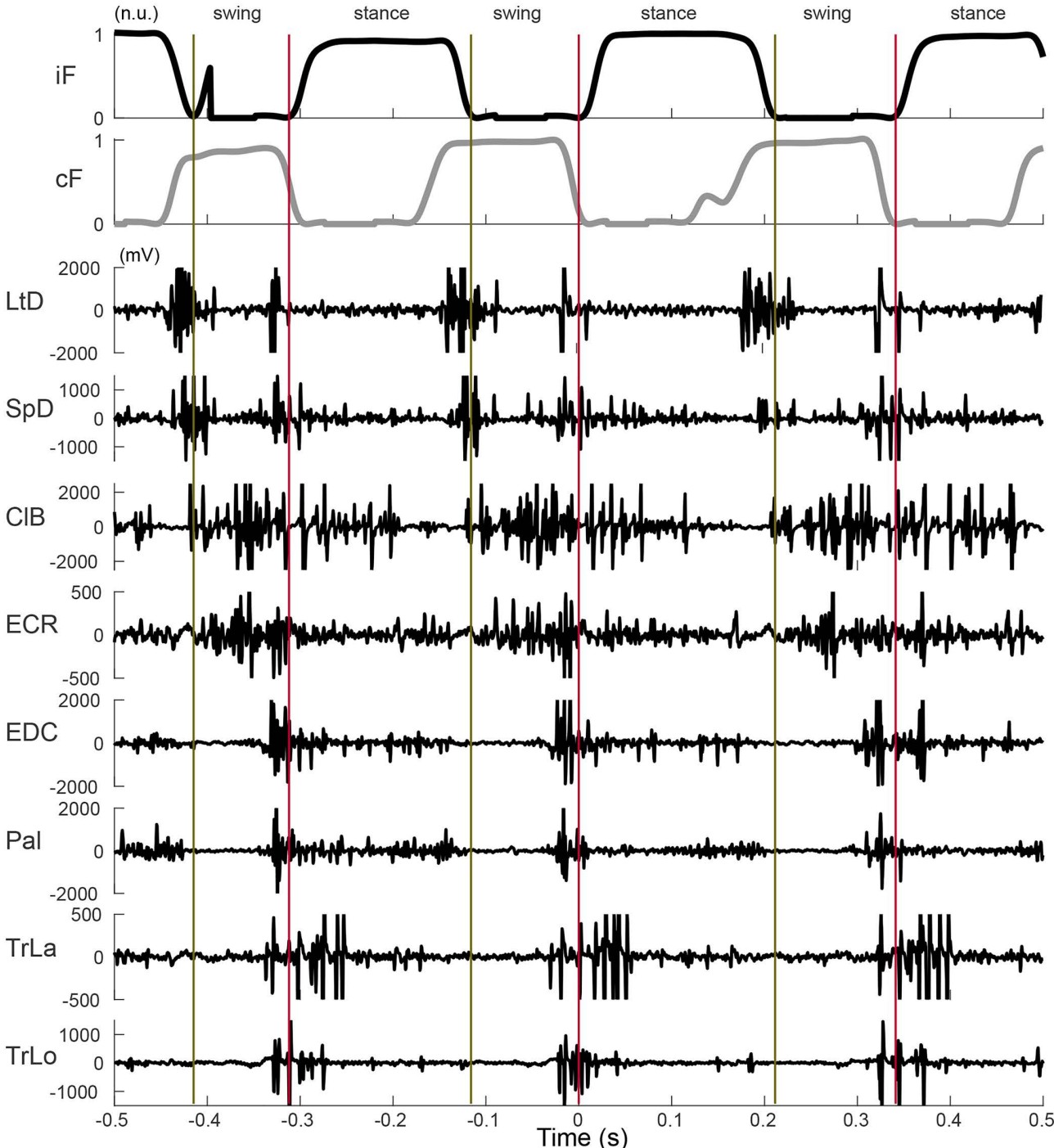

**Fig 3. Raw EMG activity and ground reaction forces during pegway walking.** Representative example of synchronized raw EMG signals from forelimb muscles and corresponding vertical ground reaction forces recorded during locomotion on the pegway. This example highlights the stereotypical temporal progression of muscle activation relative to peg stepping forces.

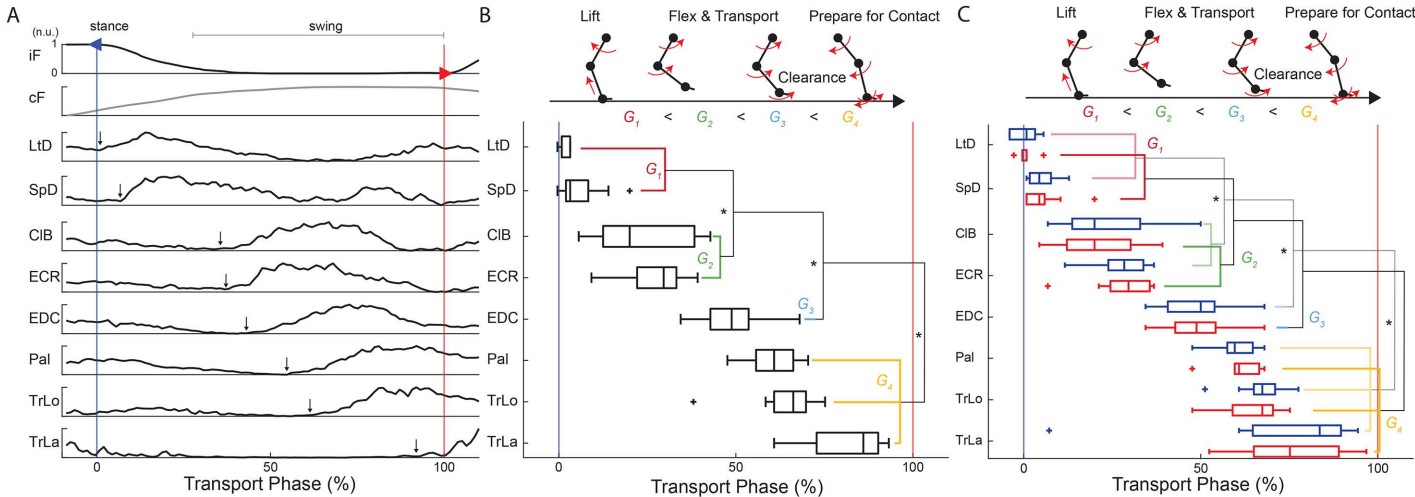

**Fig 4. Temporal progression of forelimb muscle activity during symmetric and asymmetric peg stepping. (A)** Averaged vertical ground reaction forces (GRFs, top panels) from ipsilateral (*iF*) and contralateral (*cF*) forelimbs in a representative animal performing the symmetric stepping condition (**S15**). Traces are aligned to ipsilateral limb transport onset (blue vertical line) and normalized to swing duration, with transport onset and stance onset indicated by blue and red vertical lines, respectively. Lower panels show corresponding averaged EMG profiles from implanted ipsilateral forelimb muscles, aligned to the same reference. Black arrows mark the manually detected EMG burst onsets. The sequence of muscle recruitment follows a stereotyped order, with increasing onset latencies across the swing phase. **(B)** Upper: schematic illustration of forelimb movement stages during swing phase (lift, flexion/transport, clearance, preparation for contact). Lower: box plots of EMG onset latencies for representative forelimb muscles, measured from session averages and normalized to transport onset in **S15**. **(C)** The temporal analysis is repeated for the asymmetric stepping conditions simulating ipsilateral and contralateral preference during walking (**I9C6, I6C9**). Statistical comparisons were performed using three group comparisons; significant results are indicated with (*).

Group means for ipsilateral preference (*I9C6*) (± SEM): G1 = 0.008 ± 0.011, G2 = 0.251 ± 0.027, G3 = 0.494 ± 0.037, G4 = 0.665 ± 0.034. G1 < G2: $t(5) = -7.44$, $p < 0.001$, $dz = -3.04$. G2 < G3: $t(12) = -6.49$, $p < 0.001$, $dz = -1.80$. G3 < G4: $t(12) = -7.15$, $p < 0.001$, $dz = -1.98$.

Group means for contralateral preference (*I6C9*) (± SEM): G1 = 0.004 ± 0.016, G2 = 0.256 ± 0.032, G3 = 0.491 ± 0.039, G4 = 0.659 ± 0.039. G1 < G2: $t(5) = -7.85$, $p < 0.001$, $dz = -3.20$. G2 < G3: $t(12) = -5.53$, $p < 0.001$, $dz = -1.54$. G3 < G4: $t(12) = -5.15$, $p < 0.001$, $dz = -1.43$.

EMG amplitude changes and vertical ground reaction forces (*iF, cF*) caused by an imposed asymmetry were described using a binned method (Fig 5). Signals were averaged per limb, spatiotemporally normalized, and divided into 10 bins of equal duration. Normalization of signals was done (1) for the full step cycle starting with swing onset, and (2) to the peak-to-peak values of the corresponding signal in the symmetric condition. Correspondingly, Bin 1 represents the beginning of the swing phase while Bin 10 represents the end of the stance phase. The average normalized amplitude of the portion of the signal captured in each bin was calculated, and these values are represented with box plots for each condition.

As expected, the differences in the asymmetric conditions are generally opposite, mirroring values relative to the symmetric condition. Normalization masks these differences in the ipsilateral vertical force (*iF*), and instead the promotion of limb preference is evident in the contralateral vertical force (*cF*). Using two-tailed t-test comparison within each bin, the significant differences were marked with (*) and shaded to enhance readability.

The opposite (or mirroring) effects were observed in *cF* between **I9C6** and **I6C9** conditions during the onset of ipsilateral lift (Bin 1, p < 0.001) and the ipsilateral foot contact (Bin 5 and 6, both p < 0.001). These differences, see Bins 5–8, indicate the presence of asymmetric preference across the two conditions, where contralateral support is terminated faster in **I9C6** condition (ipsilateral limb preference) as compared to **I6C9** condition (contralateral limb preference).

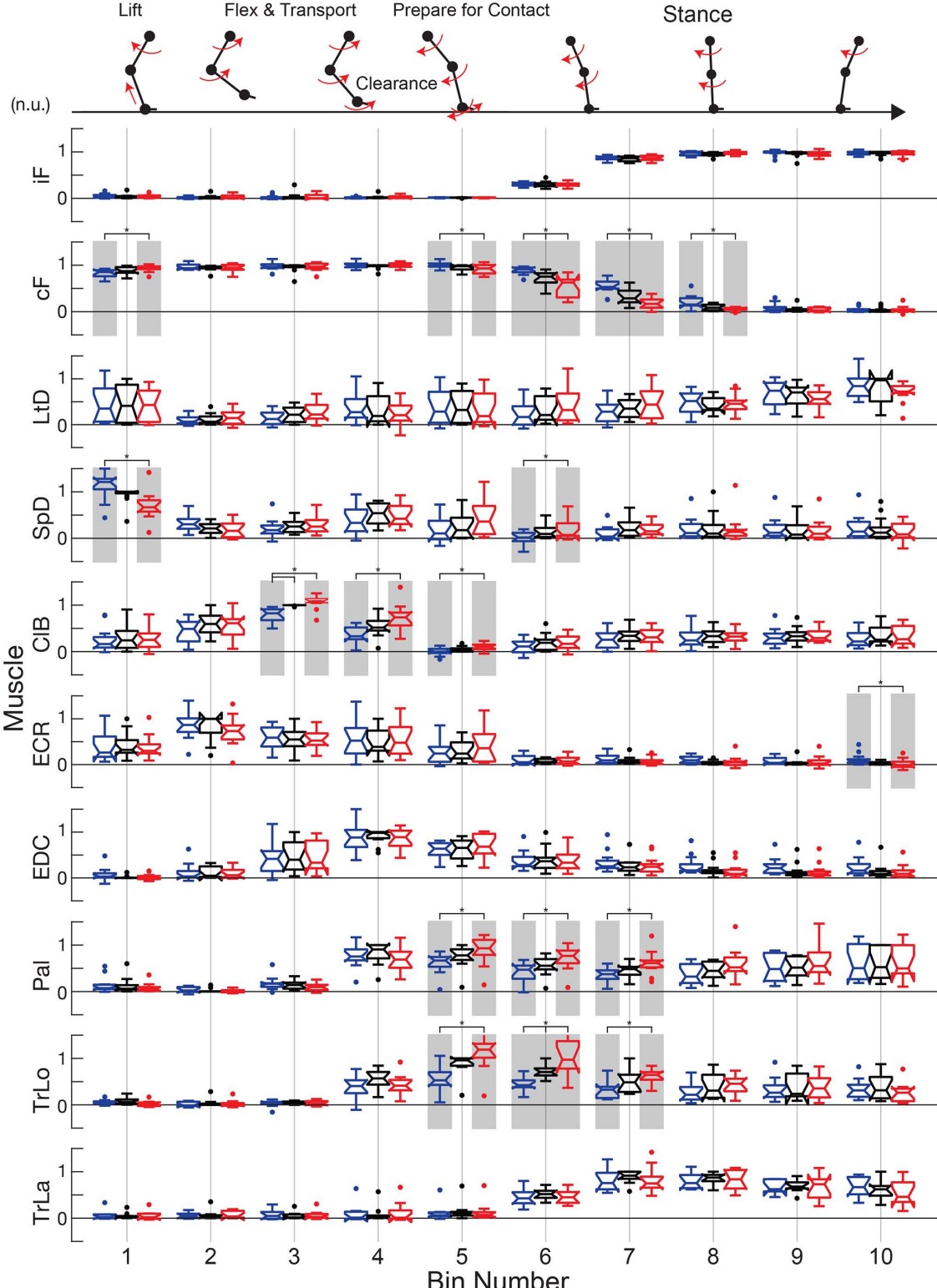

**Fig 5. Binned amplitude analysis of ground reaction force and EMG signals normalized in time to the ipsilateral limb step cycle.** Gait conditions are represented as *I6C9* in blue, *S15* in black, and *I9C6* in red. Stars and gray boxes indicate significance. The schematic kinematics in the upper region describe the movement of the forelimb during the full step cycle.

**Table 1. Muscle burst onset values for symmetric and asymmetric stepping (mean ± standard deviation is reported as a percentage of swing phase).**

| Muscle | Symmetric Stepping | Asymmetric Stepping (Ipsilateral Preference) | Asymmetric Stepping (Contralateral Preference) |
|---|---|---|---|
| LtD | 1.4 ± 1.4 | 0.4 ± 3.9 | 0.8 ± 2.7 |
| SpD | 6.1 ± 5.9 | 5.4 ± 4.0 | 5.1 ± 5.3 |
| ClB | 23.1 ± 12.7 | 23.5 ± 12.8 | 21.3 ± 11.0 |
| ECR | 26.9 ± 9.1 | 27.8 ± 7.4 | 28.9 ± 8.5 |
| EDC | 48.9 ± 8.9 | 49.1 ± 9.7 | 49.5 ± 9.1 |
| Pal | 60.5 ± 7.4 | 59.6 ± 6.2 | 61.6 ± 5.8 |
| TrLo | 64.1 ± 9.7 | 67.2 ± 6.9 | 64.4 ± 8.8 |
| TrLa | 81.5 ± 12.1 | 71.3 ± 30.3 | 76.0 ± 15.3 |

Ipsilateral EMGs exhibits the mirroring effects across the conditions. During early ipsilateral swing (Bin 1), *SpD*, limb retractor, was more active in the contralateral preference condition (*I6C9*), promoting the initiation of ipsilateral swing. *ClB*, shoulder and elbow flexor, was more active in *I9C6* than in *I6C9* during limb transfer (Bins 3–5), supporting its role in speeding the limb transfer in the ipsilateral preference condition. *EDC* showed no preference. Similarly, *Pal* and *TrLo* were supporting the faster transition to stance of the ipsilateral limb in the ipsilateral preference condition (see Bins 5–7).

### Interlimb coordination

The interlimb analysis of the double stance phase, or the period when both limbs are in stance, further characterized loco-motor asymmetry. The comparison of double stance phases and their asymmetries in different conditions are shown in **Fig 6**. Each step cycle contains two double-stance phases, which can be described as either leading or trailing. The leading double stance begins with the onset of the ipsilateral stance. It ends with a contralateral stance offset, while the trailing double stance starts with a contralateral stance onset and ends with an ipsilateral stance offset (**Fig 3**). Intralimb analyses in this study were performed per limb, while the kinematic interlimb analysis was done per animal to remove repetitions.

Double stance times were calculated using the ground reaction force 'onset' and 'peak' events in both limbs (see above *Signal Processing*). Average leading and trailing double stance times for each animal were compared within each condition using two-sided paired *t*-tests for the three within-condition comparisons. Double stance phases of leading and trailing limbs were, as expected, not different in the symmetric condition (*S15*, *t(7)*=−1.82, *p*=0.11, *dz*=−0.64). These phases were different in the asymmetric conditions (*t(7)*=5.53, p<0.001, *dz*=1.96 and *t(7)*=−5.38, *p*=0.001, *dz*=−1.902 for *I9C6* and *I6C9*, respectively), see **Fig 6A**.

Double stance time differences, calculated as trailing minus leading in each condition, were compared across condition using two-sided paired *t*-tests with the Bonferroni adjustment to maintain an overall α=0.05 across the two comparisons. The differences between the trailing and leading phases in the two asymmetric conditions exhibited mirroring values relative to the symmetric walking, **Fig 6B** (**I6C9-S15**: *t(7)*=−10.57, p<0.001, *dz*=−3.736; **S15-I9C6**: *t(7)*=−6.90, p<0.001, *dz*=−2.44) indicating consistent modulation of the corresponding subphases across the conditions.

### Discussion

We demonstrate that rats exhibit a stereotyped sequence of forelimb muscle activation during both symmetric and asymmetric precise stepping, closely paralleling patterns described in cats during locomotion and reaching. Previously, the pattern of sequential muscle activation has been described in cats during locomotion on a treadmill with and without obstacles, as well as during reaching movements to a lever [72,73]. While kinematic analysis of stepping in rodents has been extensively based on motion tracking and even exoskeleton use [74–76], the description of muscle coordination

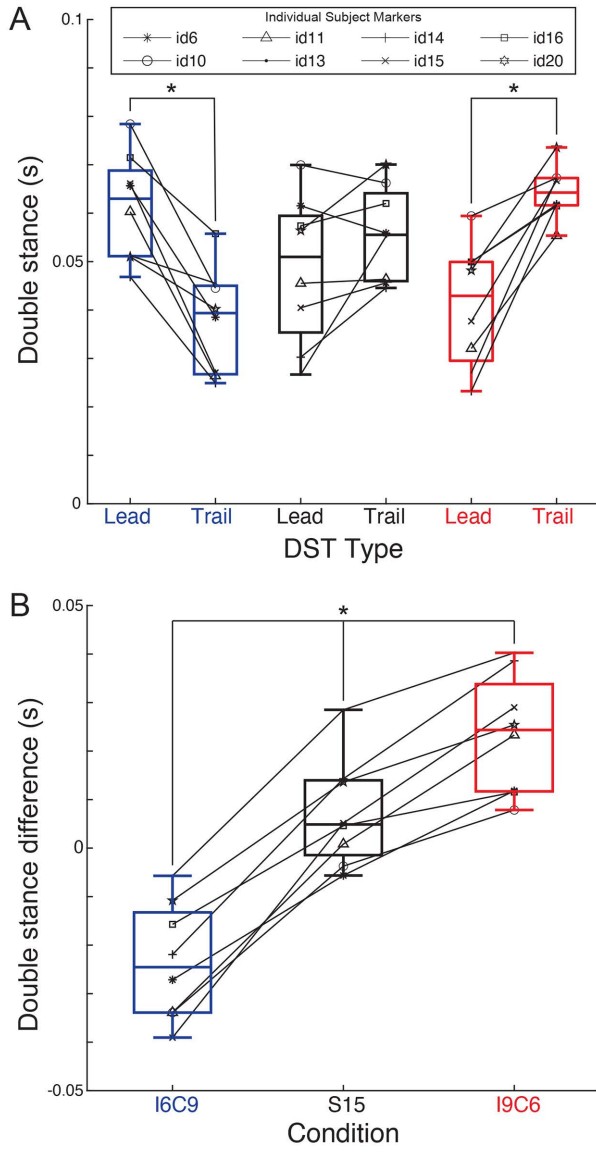

**Fig 6. Double stance analysis.** Subjects are indicated with unique markers; black diagonal lines indicate intra-subject changes in double stance times. **(A)** Separated leading and trailing double stance times for each condition (*I6C9*, *S15*, *I9C6*) and each subject. **(B)** Double stance time differences (trailing minus leading) in each condition and each subject. Significant results are indicated with (*).

during symmetric and asymmetric precise stepping is a novel result in this study. A stereotyped spatiotemporal sequence of activation in representative forelimb muscles was persistent and mirroring across the asymmetric tasks.

## Muscle sequence for symmetric and asymmetric gait

Rats use a similar progression of muscle activity as cats [3,20] to execute limb lift, flexion, transport and preparation for stance phase of locomotion (Fig 4 and 5). The muscle recruitment onsets are organized in the following sequence: limb retractors (*LtD, SpD*), elbow flexors (*ClB, ECR*), wrist dorsiflexor (*EDC*), followed by plantar-flexor and load bearing muscles during stance (*Pal,TriLo, TriLa*). The burst durations of *ClB* and *ECR* were similar (**Fig 4A**), unlike in cats [73], and

likely associated with the requirement to maintain flexion for ground clearance in the crouched posture of rats. Furthermore, the differences observed between cats and rats at the elbow could be described by differences in passive dynamics. For example, during swing phase in cats, the knee joint extends passively due to inertia of the shank and foot while the relatively smaller limb of rats require force generation through muscle activity [77,78]. While cortical activity during precise stepping in rats has not been reported, a sequential pattern of cortical discharge has been observed in cats and was associated with the coordination of muscles during reaching movements [73] and also during locomotion [20]. We hypothesize that similar cortical dynamics are responsible for the conservation of spatiotemporal muscle activation across cats and rats. However, the slight difference in muscle bursting as seen in *CIB* and *ECR* suggests the need for further investigation.

Complex interactions between multiple feedforward and feedback as well as spinal and supraspinal pathways provide online control of stepping and contribute to muscle activations [79]. However, the reason for the expression of integrated patterns at the cortical level is unknown. The contribution of descending control inputs is essential for voluntary gait modifications and precise stepping, evidenced by the compromised postural control in decerebrated cats despite the fact that their sensory pathways remain intact [21]. Previously, the existence of these persistent behavioral templates at multiple levels in the control hierarchy was attributed to the challenging task of integrating limb and whole-body posture control [20]. This hypothesis is supported by the view of internal representations of limb and body dynamics necessary for the planning and execution of movement in the presence of transmission delays and unpredictable interactions with the environment [80,81] and with the potential need for sensorimotor adaptations [82]. While this holistic view on control suggests complex representation of the whole system at the high levels in the hierarchy, it includes the additive layered organization of multiple feedback loops. The output of such a system is an emergent property of the interactions between descending and ascending pathways.

## Low-dimensional and high-dimensional representation of control

Spatiotemporal templates of muscle activity that can be independently modified contradict the notion of a purely low-dimensional control scheme. Evidence for low-dimensional representations often comes from dimensionality reduction studies in reaching, posture, and locomotion [83,84]. Such methods, like principal component analysis, can mathematically compress almost any pattern into a few components [85–87]. The models of pattern generation also predict that simple limb speed signals can account for the appropriate speed-dependent phase modulation and changes in heading direction during turning (Yakovenko et al., 2018). In this view, low-dimensional inputs such as limb speed are transformed by spinal circuits into high-dimensional outputs that recruit many muscles [88] and can be integrated with supraspinal inputs [89], see **Fig 7**. Postural and movement control can also be simplified into "anchor behaviors," like the spring-loaded inverted pendulum, which serve as building blocks for predictive and reactive neuromechanical control [90]. However, low-dimensional structure does not exclude the presence of high-dimensional descending signals. Cortical neurons show phasic tuning to muscle activity during locomotion, with contributions that vary depending on task demands [13,91]. These cortical inputs become critical for precision and dexterity [92]. Walking across a ladder, for example, requires corticospinal involvement, and lesions in this pathway disrupt performance in such tasks.

Low-dimensional representations such as CPGs do not imply that the nervous system relies exclusively on low-dimensional computations. When tasks demand dexterity, high-dimensional control can be recruited alongside these representations to expand the behavioral repertoire. For instance, locomotion can be combined with precise stepping over obstacles, which requires detailed force representations during swing subphases or reaching movements [21,73]. **Fig 7** illustrates the concept where dynamic computations associated with specific groups are coupled to the oscillatory regulation of the CPG, but can be independently controlled by inputs from multiple sources—descending inputs, other intrinsic CPG projections, or sensory feedback.

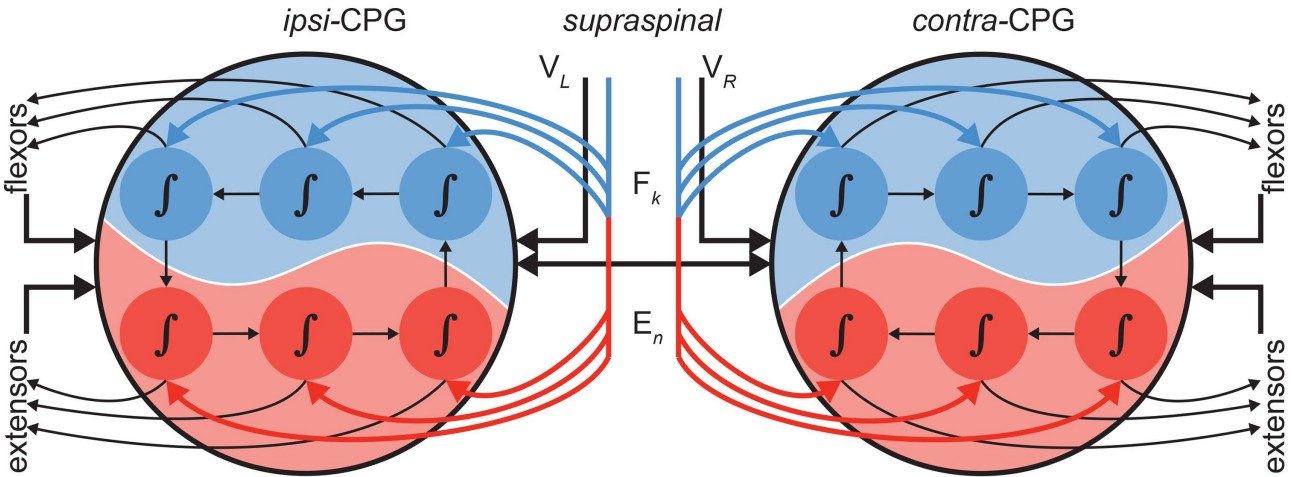

**Fig 7. Descending control integrates with central pattern generator (CPG) dynamics to shape sequential activation of neuromechanical groups.** The schematic illustrates how supraspinal inputs modulate ipsilateral and contralateral CPG circuits, which in turn coordinate flexor and extensor muscle groups in symmetric and asymmetric locomotor tasks. This sequential organization underlies whole-limb control during rhythmic movement.

Our findings indicate that asymmetric locomotor conditions modulated specific muscle groups supporting the idea that the dynamic computations for the associated groups were embedded within and coupled to the oscillatory regulation of the CPG. These dynamic modules—such as those driving flexors and extensors—receive rhythmic timing from the CPG but remain independently modifiable, see **Fig 7**. Control signals can arrive from multiple sources, including descending supraspinal inputs [see also 90], propriospinal projections from other CPG elements [93,94], or peripheral sensory feedback [95]. This organization allows for a stable oscillatory backbone that ensures variable locomotion, while simultaneously providing the flexibility to adjust coordination in response to environmental demands. By combining these layered mechanisms, the system can preserve robust locomotor patterns with precision and adaptability, for example, during obstacle negotiation or asymmetric stepping.

The present findings support the idea that spatiotemporal templates underlie such dexterous movements, as shown by the similarity in the temporal progression of muscle activity across species. These templates can be captured in a low-dimensional structure, yet the same control pathways that place the limb accurately in symmetric walking also adapt to asymmetric conditions by modifying a shared motor program. Importantly, the balance between mechanisms appears flexible, shifting with the demands of the task [29]. Cross-species comparisons reinforce this view. Both rats and cats exhibit similar temporal muscle activity patterns during locomotion [12,20,22,96] and reaching [20,73,97]. The synergy analyses using PCA and non-negative matrix factorization likewise reveal consistent spatiotemporal activity templates across species [98]. Taken together, these results point to a conserved organization of motor coordination in both rat and cat models.

## Conclusions

This study provides the first detailed characterization of forelimb muscle activity during precise stepping in rats. The spatiotemporal sequence of activation was highly stereotyped and closely paralleled patterns previously described in cats during both reaching and locomotion with obstacles. This cross-species similarity underscores the existence of conserved spatiotemporal templates in quadrupedal motor control. Our findings highlight that rats, like cats, recruit sequentially organized muscle groups to achieve dexterous limb placement in both symmetric and asymmetric stepping. This conservation suggests that the neural mechanisms underlying intralimb coordination are generalizable across species, despite differences in morphology and posture. Importantly, the results establish rats as a reliable model for studying the neural control

of skilled locomotion, bridging basic neurophysiological principles in larger quadrupeds with the accessibility and versatility of rodent models. By documenting these templates, we provide a foundation for future studies examining cortical and spinal contributions to precise locomotor adjustments, and we strengthen the translational relevance of rat models for understanding motor disorders and developing rehabilitative strategies.

## Acknowledgments

We would like to acknowledge the assistance of Sarah Freeman in animal care, training, data collection, and surgeries.

## Author contributions

**Conceptualization:** Ezequiel M. Salido, Kiril Tuntevski, Sergiy Yakovenko.

**Data curation:** Kacie Hanna, Ezequiel M. Salido, Neha Lal, Kiril Tuntevski, Sergiy Yakovenko.

**Formal analysis:** Kacie Hanna, Ezequiel M. Salido, Neha Lal, Kiril Tuntevski, Sergiy Yakovenko.

**Funding acquisition:** Sergiy Yakovenko.

**Investigation:** Ezequiel M. Salido, Sergiy Yakovenko.

**Methodology:** Kacie Hanna, Ezequiel M. Salido, Neha Lal, Kiril Tuntevski, Sergiy Yakovenko.

**Project administration:** Sergiy Yakovenko.

**Resources:** Neha Lal, Sergiy Yakovenko.

**Software:** Kacie Hanna, Sergiy Yakovenko.

**Supervision:** Sergiy Yakovenko.

**Validation:** Sergiy Yakovenko.

**Visualization:** Kacie Hanna, Sergiy Yakovenko.

**Writing – original draft:** Kacie Hanna, Sergiy Yakovenko.

**Writing – review & editing:** Ezequiel M. Salido, Neha Lal, Kiril Tuntevski, Sergiy Yakovenko.

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
