## [Decision Letter · Decision Letter 0]

9 Dec 2025

Spatiotemporal forelimb muscle activation in rats during asymmetric peg stepping

PONE-D-25-49468

Dear Dr. Yakovenko,

We’re pleased to inform you that your manuscript has been judged scientifically suitable for publication and will be formally accepted for publication once it meets all outstanding technical requirements.

Kind regards,

Carlos Tomaz, Ph.D.

Academic Editor

PLOS ONE

Additional Editor Comments (optional):

The manuscript was carefully reviewed by two reviewers, and both concluded that the manuscript can be accepted for publication in its current form. I concur with the reviewers' opinion.

Reviewers' comments:

Reviewer's Responses to Questions

**Comments to the Author**

1. Is the manuscript technically sound, and do the data support the conclusions?

Reviewer #1: Yes

Reviewer #2: Yes

2. Has the statistical analysis been performed appropriately and rigorously?

Reviewer #1: Yes

Reviewer #2: Yes

3. Have the authors made all data underlying the findings in their manuscript fully available?

Reviewer #1: Yes

Reviewer #2: Yes

4. Is the manuscript presented in an intelligible fashion and written in standard English?

Reviewer #1: Yes

Reviewer #2: Yes

Reviewer #1: Dear Editor-in-Chief and authors,

My greetings.

The article entitled “Spatiotemporal forelimb muscle activation in rats during asymmetric peg stepping” presents an investigation of the spatiotemporal sequence of forelimb muscle activation in rats performing a precise locomotor task — walking across an array of pegs under symmetric and asymmetric stepping conditions.

The goal stated in the Introduction was achieved through a carefully designed experimental procedure using intramuscular electrodes implanted in groups of forelimb muscles of Sprague–Dawley rats. The results corroborate previous findings in cats and also provide new insights; therefore, the research hypothesis was successfully addressed.

In general, the article is written in an appropriate scientific style and demonstrates theoretical consistency, generating data that may benefit future studies aimed at understanding the precise muscular coordination involved in limb movement and, indirectly, the cortical pathways responsible for such control. These findings could also contribute to clinical applications in human physiotherapeutic management.

In my opinion, the article deserves to be accepted for publication.

Some additional information could be included, but I prefer to respect the authors’ creativity and writing style.

Reviewer #2: This manuscript presents data derived from an original study, characterized by a meticulously described experimental design and a high level of detail. The data exhibit statistical significance.

It is suggested that minor revisions be implemented, specifically incorporating more contemporary references in the introduction and discussion sections.

This research is an important and highly significant foundation for subsequent research examining cortical and spinal contributions.

**Do you want your identity to be public for this peer review?** For information about this choice, including consent withdrawal, please see our Privacy Policy

Reviewer #1: **Yes: ** Tales Alexandre Aversi Ferreira

Reviewer #2: No

---

## [Editor Report · Acceptance letter]

PONE-D-25-49468

PLOS One

Dear Dr. Yakovenko,

I'm pleased to inform you that your manuscript has been deemed suitable for publication in PLOS One. Congratulations! Your manuscript is now being handed over to our production team.

Kind regards,

on behalf of

Dr. Carlos Tomaz

Academic Editor

PLOS One